# Influence of Carbon Sources on the Phenolic Compound Production by *Euglena gracilis* Using an Untargeted Metabolomic Approach

**DOI:** 10.3390/biom12060795

**Published:** 2022-06-07

**Authors:** Eve Bernard, Céline Guéguen

**Affiliations:** Département de Chimie, Université de Sherbrooke, 2500 Boulevard de l’Université, Sherbrooke, QC J1K 2R1, Canada; eve.bernard@usherbrooke.ca

**Keywords:** Folin–Ciocalteu method, glutamic acid, metabolites, polyphenol, euglenoids

## Abstract

Industrial development and urbanization has led to the diverse presence of metals in wastewater that are often improperly treated. The microalgae *Euglena gracilis* can tolerate high concentrations of metal via the excretion of organic metabolites, including phenolics. This study aims to evaluate how carbon amendment stimulates phenolic compound production by *E. gracilis*. The number, relative intensity and molecular composition of the phenolic compounds were significantly different between each of four carbon amended cultures (i.e., glutamic acid, malic acid, glucose, reduced glutathione) during the log phase. Phenolic compounds were mainly produced during the minimum growth rate, likely a response to stressful conditions. A better understanding of phenolic compounds production by *E. gracilis* and the impact of growth conditions will help identify conditions that favor certain phenolic compounds for dietary and metal chelation applications.

## 1. Introduction

Metal presence in aquatic ecosystems is an increasing global problem largely due to rapid urbanization and industrialization [1]. An estimated 40% of lakes and rivers has already been contaminated [2]. Metals can cause adverse and hazardous effects on living organisms, vegetation, biodiversity and human health [3,4]. Therefore, it is important to find a way to properly treat industrial and municipal effluents. Several physical, chemical, and biological methods have been developed to this end. One of them is phycoremediation, a form of bioremediation, using algal biomass producing chelating ligands able to recover significant amounts of metals [5].

*Euglena gracilis* is a unicellular aquatic organism part of the euglenoids family. This free-floating protist is able to produce energy under autotrophic, heterotrophic and mixotrophic conditions [6] by utilizing carbon from several sources [7,8]. Its metabolic flexibility suggests the existence of several physiological pathways affecting the biosynthesis and proportion of different metabolites [9]. *Euglena gracilis* also shows high adaptation capacities for a broad range of environmental conditions [10] through the plasticity of its atypical and unique metabolism [6,11], including its ability to tolerate high concentrations of metals (ppm range) [12] via the production of chelating ligands [13]. These studies focused mainly on primary metabolites production, but little is known in terms of secondary metabolites [14]. Secondary metabolites are specialized compounds produced in response to environmental changes [15]. They play a key role in the defense mechanism against abiotic and biotic stress via their anti-allergenic, anti-microbial, anti-inflammatory and antioxidant properties [16,17]. For example, oxidative stress led to the formation of oxidative compounds and free radicals, and the breakdown of photosynthetic and metabolic enzymes [14,18].

Phenolic compounds are a major class of secondary metabolites and act as reducing agents and hydrogen donors to minimize oxidative stress [19,20]. In addition, phenolic compounds show an excellent metal chelating potential due to the abundance of oxygen functional groups in their structure [21,22]. These specialized metabolites are synthesized because of interactions with the environment [23] and can reflect evolutive environmental conditions, such as nutrient availability or abiotic stress presence. Altered growth and medium conditions can promote the production of specific metabolites [18,24,25]. For example, metabolites with carboxyl functional groups were more abundant in *Bacillus subtilis* cells harvested during the exponential phase than during the stationary phase [26]. Other studies exploring the effects of changing growth light regimes found a significant effect on metabolite composition and the production of metal binding ligands [27,28]. Few reports have focused on phenolic compounds and their bioactive functions [29] but their molecular composition remains largely unknown.

The aim of this study was to assess how the production of phenolic compounds by *Euglena gracilis* cultures is influenced by the carbon source (i.e., glutamic acid, malic acid, glucose or reduced glutathione). Earlier studies [30,31,32,33] have shown the direct and important impact of growth conditions, as variable organic carbon sources have an impact on secondary metabolite production. The most common analytical method for the quantification and characterization of phenolic compounds is UV-visible spectrophotometry using the Folin–Ciocalteu (FC) reagent method [34]. The FC assay is a rapid, repeatable method to measure total phenolic content and oxidative capacity of algal cells, but phenolics, proteins and thiols are also reactive to the FC reagent [35,36]. High performance liquid chromatography (HPLC) combined with mass spectrometry (MS) has been shown to be suitable for the detection of phenolic compounds [37,38]. Here we describe an untargeted metabolomic approach to unravel, for the first time, the molecular composition of the cellular phenolic compounds produced by *Euglena gracilis* as a response to different carbon sources.

## 2. Materials and Methods

### 2.1. Algal Growth and Biomass Harvesting

*Euglena gracilis* culture was obtained from the Canadian Phycological Culture Centre (CPCC; University of Waterloo, Waterloo, ON, Canada) and grown in modified acid medium (MAM) [39] at pH 3 and supplemented with one simple carbon source, i.e., glutamic acid, malic acid, glucose or reduced glutathione (GSH) at a concentration of 5 g L^−1^. Cultures were grown in a pre-combusted 1L Erlenmeyer flask and maintained under photoautotrophic conditions with an alternating light:dark cycle of 16:8 h. The light intensity was fixed at 150 µmol photons m^−2^ s^−1^ and the temperature kept between 20–25 °C. Previous studies [40,41,42] have shown that the addition of simple carbon sources in photoautotrophic conditions allowed faster growth. Having a greater biomass was crucial to carry out an in-depth study of the molecular characterization of phenolic compounds produced by *E. gracilis*. The cell concentration was counted using a Leica DM500 light microscope and a hemacytometer. The initial cell density of each culture was 1.0 × 10^6^ cell mL^−1^, and the biomass was harvested on day 1, 3, 5, 6, 8 and 10 of growth. The cell pellet was washed 3 times with milliQ water and stored freeze-dried until further analysis.

### 2.2. Antioxidant Activity Assay

The antioxidant activity was measured with the FC assay [43]. Briefly, 10 mg of dry biomass was ground and sonicated for 1 h in 85% methanol then centrifuged for 30 min at 4900 rpm. The FC reagent (1:10) was added to the sample extract or standard. After a 5 min incubation at room temperature, 1.6 mL 1 M Na_2_CO_3_ was added, and the reaction mixture incubated for 2 h in the dark at room temperature. The absorbance was measured at 760 nm with a spectrophotometer (Shimadzu UV-1800). The antioxidant activity was expressed in gallic acid equivalent (mg g^−1^ dry weight) using a calibration curve ranging from 0 to 62.5 µg mL^−1^.

### 2.3. Phenolic Compounds Extraction and HPLC Separation

Forty mg of biomass was extracted with polar solvents (water, methanol and ethanol), 2,3-ter-butyl-4-hydroxyanisol (2 g L^−1^) and formic acid (0.1%) using a sonic bath for 1 h [44,45]. The choice of solvent (20, 50, 80% of methanol:MilliQ or ethanol:MilliQ) was dependent on the polarity of phenolic compounds in *E. gracilis* biomass. After centrifugation at 4900 rpm for 30 min, the supernatant was collected and evaporated using a rotary evaporator at 30 °C. The residues were resuspended in methanol (20%) and acidified water (1% formic acid) and filtered using a 0.2 µm polyesthersulfone filter. Phenolic compounds were then separated by gradient elution high performance liquid chromatography (HPLC). The chromatographic separation was performed on an Agilent system with a vacuum degasser, a binary pump, a thermostat column compartment, and a diode array detector. A reversed-phase Poroshell 120 EC-C18 column (4.6 × 100 mm; 2.7 µm) was used. The mobile phase flow rate was 0.6 mL min^−1^ and the column was kept at 30 °C. The two mobile phases used were HPLC grade water/formic acid (99.9:0.1 *v*/*v*; eluent A) and HPLC grade methanol/formic acid (99.9:0.1 *v*/*v*; eluent B). The gradient profile was described as follows: isocratic 20% B (0–1 min), 20–60% B (1–8 min), 60% B (8–10 min), 60–100% B (10–12 min), isocratic 100% B (12–15 min), 100–20% B (15–16 min), isocratic 20% B (16–20 min). Nine standards (SigmaAldrich, Millipore Sigma, Oakville, ON, Canada) were used for the chromatographic separation: gallic acid, catechin, chlorogenic acid, caffeic acid, p-coumaric acid, ferulic acid, naringenin, quercetin and kaempferol. Five different fractions were collected at 4.7, 6.5, 7.1, 10.5 and 11.3 min for MS analysis.

### 2.4. Mass Spectrometry Analysis

The metabolites associated with each of the five LC fractions were analyzed using a MAXIS time-of-flight mass spectrometer (qTOF; Bruker Daltonics, Bremen, Germany) equipped with an electrospray ionization (ESI) source in negative ionization mode. The samples were directly infused at a flow rate of 300 µL h^−1^ (Lewis et al., 2021). The signal was acquired for 2 min across a *m*/*z* range of 100–1000. The source capillary voltage was set to 5000 V with a nitrogen gas flow rate of 4 L min^−1^ and a capillary temperature of 180 °C. An external standard (sodium formate) was used to ensure a good calibration from day to day. MS grade methanol was infused between each sample to minimized sample carry over and contamination. A blank sample was also acquired every 10 samples and all *m*/*z* found in the blank were removed from sample spectra to remove background. Spectra processing was performed using Compass DataAnalysis (v4.4, Bruker Daltonics, Bremen, Germany) where a mass list was generated at S/N > 4 and a relative intensity cut-off of 0.1%. The elemental composition was then attributed using SmartFormula (DataAnalysis v4.4) with the following criteria: ^12^C(1-50), ^1^H(1-100), ^16^O(1-30), ^14^N(0-2), ^32^S(0-2) within 10 ppm mass error. We confirmed the molecular formulas for the most abundant peaks with the help of the signal intensities of ^13^C_1_^12^C*_n_*_−1_ compounds. Only the *m*/*z* present in replicate samples were kept for phenolic analysis. Here, phenolic compounds were defined as 0.6 < H/C < 1.5 and 0.3 < O/C < 0.85 [46,47] and a modified aromaticity index (AImod) inferior to 0.67 [48].

### 2.5. Statistical Analysis

The different growth phases were identified for each carbon source using a curve fitting model (GraphPad Prism 9). Each carbon source had 4 days corresponding to the log phase except for glucose, which had only 3 days. During those days, we were able to identify the maximum slope and the minimum slope using the growth rate equation for a population as follows:$$(µ)=\frac{\ln N_{2}-\ln N_{1}}{t_{2}-t_{1}}$$

A Shapiro–Wilk normality test was performed followed by the Wilcoxon nonparametric test (R studio). A significant difference was considered at the level of *p* < 0.05.

## 3. Results

### 3.1. Growth Curve and Carbon Sources

The MAM medium was supplemented with simple carbon sources to achieve faster growth and a higher cell density relative to inorganic media (Appendix A). Each growth medium presented a different profile because each carbon source is metabolized differently by *E. gracilis* favoring specific and variable metabolic pathways that impact on growth. The length of the log phase was also dependent on the carbon source (Figure 1). Glutamic acid and its amine group supports the growth of cells via its ability to biosynthesize amino acids and nucleic acids. Glucose is the main carbon source in cells and the elemental metabolite for glycolysis, which helps produce energy and many molecules of biological interest. Malic acid is involved in the tricarboxylic acid cycle (TAC) [49], which is responsible for the majority of energy production in cells. These three carbon sources have a direct impact on energy production and growth as depicted by a significantly higher biomass compared to the inorganic amended culture. In comparison, the GSH amendment showed a smaller increase in cell density compared to the other C sources but it was higher than MAM alone. GSH is a major endogenous antioxidant responsible for the redox balance management and is involved in the detoxification of xenobiotic and endogenous compounds in cells [50].

### 3.2. Phenolic Concentration Based on the Folin–Ciocalteu Assay

The concentration in gallic acid equivalent (GAE) of the extracted phenolic compounds varied from 2.15 to 2.46 mg g^−1^ and from 1.44 to 2.11 mg g^−1^ in the methanol and ethanol extracts, respectively (Appendix A). More phenolic compounds were found in the methanol extracts than in the ethanol extracts (*p* < 0.05), suggesting that the phenolic compounds in *E. gracilis* were highly hydrophilic. Previous studies reported better phenolic extraction with organic aqueous mixtures [51] and a better extraction yield using methanol [50]. Overall, both the 50% and 80% methanol extracts showed the highest concentrations with values of 2.39 ± 0.04 and 2.46 ± 0.02 mg g^−1^, respectively. Further analyses were performed using the 80% methanol extracts.

The mean antioxidant capacity using the FC assay ranged from 4.04 ± 0.52 mg g^−1^ with malic acid amendment to 6.08 ± 0.73 mg g^−1^ with GSH amendment (Figure 2). Comparable values have been previously reported for *Euglena gracilis* and other microalgae [52,53]. The mean antioxidant capacity was 1.2–1.5 times higher with the sulfur-rich amendment (GSH amendment) than with any other carbon amendments. This result was expected as sulfur has a good antioxidant capacity [54].

### 3.3. Phenolic Compound Molecular Characterization

The reverse-phase HPLC method presented good resolution and reproducibility for the 9 different standards tested (Appendix A) and good linearity over the concentration range (12.5–200 µM; Appendix A). None of the pure standards was found in the algal samples isolated in this study (Appendix A). The elemental ratios (O/C and H/C), the modified aromaticity index (AImod), double bond equivalent (DBE), and the nominal oxidation state of carbon (NOSC) were calculated (Appendix A) for the five fractions F1–F5 (Appendix A) obtained from chromatographic separation of the cellular extracts. No significant trends were observed between fractions, highlighting the heterogeneity of the phenolic compounds isolated in the chromatographic fractions. The molecular composition was performed on the combined fraction only (F1 + F2 + F3 + F4 + F5).

The number of phenolic compounds varied greatly from day to day during the log phase between C sources (Figure 3A–D). A range of 6–31 different *m*/*z* attributable to phenolics were found with no significant differences between carbon sources (*p* > 0.05). Their relative intensities varied between C sources (0.057–0.296; Figure 3E–H) with the glutamic acid amendment showing the greatest variability.

## 4. Discussion

### 4.1. Impacts on Phenolic Production

Significant differences in phenolic production were found, depending on the cellular growth rate. The lowest number and relative intensity of phenolic compounds in the glutamic acid amendment (Figure 3A,E) were found on day 3 of growth, which represents the middle of the log phase and the start of the maximum growth rate (Figure 1A). Glutamic acid supports cell growth by producing nitrogen-containing compounds. The variation in phenolics and the minimum production of phenolic compounds when cell division is the most important may imply that glutamic acid is used preferentially for other essential biological functions when the cell division rate is maximal and is more favorable for phenolic production when the cell is less metabolically active. The glucose amendment (Figure 3B,F) showed a relatively similar pattern, with a decrease in the number of phenolic compounds as the growth curve slope steepened. Glucose is used directly by the cells to produce energy and different biological compounds. Although glucose is involved in the production of phenolic compounds via the synthesis of phosphoenolpyruvate [55], it has many additional functions. The malic acid amendment (Figure 3C,G) had the lowest overall relative intensity relative to the other carbon sources, likely due to its involvement in the TAC and the production of adenosine triphosphate (ATP). The cellular energy demand throughout the log phase is relatively steady, which means that malic acid was constantly used to provide energy to support growth rather than to produce phenolic compounds. The small overall increase in phenolic compounds production during the log phase may be explained by ATP production leading to the activation of different secondary metabolic pathways, including those for the synthesis of phenolic compounds. The phenolic compounds in the GSH amendment (Figure 3D,H) were maximal at mid-exponential phase. Metabolic waste and toxins accumulate as the log phase progresses, and since the GSH is used for detoxification purposes [50], this can explain the diminution at the end of the growth phase. The relative phenolic intensity decreased as the growth slope slightly increases with the glutamic acid, malic acid, and glucose amendments.

The phenolic compounds produced during the log phase changed rapidly from day to day. Up to 11 common *m*/*z* (corresponding to 9–24% of assigned *m*/*z*) were found between two consecutive days in each carbon source (Appendix A), highlighting the rapid transformation of phenolic compounds produced by *E. gracilis*. Most of the assigned formulas were unique to a carbon source, indicating a significant impact of the carbon source on the phenolic compounds profile.

Phenolic compounds with significantly higher *m*/*z* were found at the end of the log phase in comparison to the beginning for glucose and GSH (Appendix A). This is consistent with the fact that the cell is the most metabolically active during this growth stage and has enough time and resources to biosynthesize more complex molecules, which often requires more energy and more sophisticated biochemical pathways.

### 4.2. Impacts on Phenolic Molecular Composition

The growth rates showed significant differences in the phenolic profiles (Appendix A). The maximum growth rate was characterized by a lower number and relative intensity of phenolic compounds for each carbon amendment except for the number of phenolic compounds in the glucose amendment. Phenolic compounds have antioxidant properties, and as such, are more prone to be produced when the cells are stressed. During the maximum growth rate, the cells are under optimal conditions for metabolic activity and growth, in agreement with the observed reduction in the production of phenolic compounds.

The elemental composition of the phenolic compounds showed significant differences between minimum and maximum growth rates (Figure 4). The elemental composition for each carbon source was more balanced during the minimum growth rate with a particularly comparable proportion of CHO (26–36%), CHON (26–29%) and CHOS (12–26%) (Figure 4A). These results contrast with the maximum growth rates where the elemental composition was more variable (Figure 4B). A certain level of stress was present during the minimum growth rate that can influence the production of phenolic compounds which can cope with the stress. In contrast, during the days corresponding to the maximum growth rate, the stress was minimal, and the carbon source was used to support cellular growth and metabolic activity, which greatly affects the phenolic elemental profiles.

Malic acid contains no nitrogen or sulfur, consistent with the very low CHON (22%), CHOS (6%) and CHONS (19%) abundances in the produced phenolics with the malic acid amendment. The abundance of CHO phenolic compounds accounted for 53% of the phenolic compounds. The presence of sulfur in GSH was translated into S-rich phenolic compounds. Up to 71% of the phenolic compounds identified contained sulfur (Figure 4B). The elemental composition of the phenolic compounds produced with the glutamic acid and glucose amendment were comparable, with CHO dominating the phenolic formula at both growth rates (*p* > 0.05; 26–36%; Figure 4B). No distinct heteroatom rich phenolic compounds were found in the glucose amendment, likely due to the fact that glucose is the primary energy source for cellular activity and is used to produce a great variety of biological compounds [56]. A similar explanation can justify the absence of a distinct profile with glutamic acid treatment, since it is used for a great diversity of N-rich molecules, including nucleic acid and amino acid biosynthesis [57]. The glutamic acid treatment did not show a higher abundance of CHON and CHONS since nitrogen is mainly used in the production of biomass [58]. Together these results showed that the number, the intensity, and the elemental composition of cellular phenolic compounds were dependent on the growth rate of *E. gracilis* and the carbon source present.

## 5. Conclusions

This study focuses on the influence of four simple carbon additions on the production of phenolic compounds by *Euglena gracilis* cells using an untargeted metabolomic approach. The antioxidant capacity of *E. gracilis* cellular extracts determined by Folin–Ciocalteu assay was highly variable between carbon sources during the log growth phase. The cellular phenolic compounds were highly diversified in number, relative intensity and molecular composition between the four carbon treatments and during the log phase of *E. gracilis* growth. The minimum growth rate period showed up to a 3-fold increase in cellular phenolic production compared to the optimal growth rate, suggesting that cellular phenolic compounds were mostly produced to mitigate stressful conditions. The predominance of sulfur- and nitrogen-rich phenolic compounds was 7–14% higher during the minimum growth rate. Favorable conditions for the production of phenolic compounds could be particularly advantageous in dietary supplement and metal chelation applications.

## Figures and Tables

**Figure 1 biomolecules-12-00795-f001:**
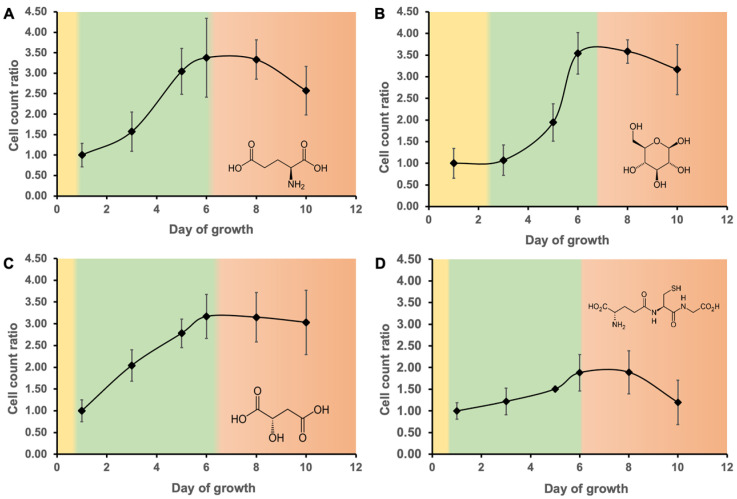
Growth curve and the different phases with (**A**) glutamic acid, (**B**) glucose, (**C**) malic acid and (**D**) GSH amendment during lag (yellow), log (green), and stationary phases (red).

**Figure 2 biomolecules-12-00795-f002:**
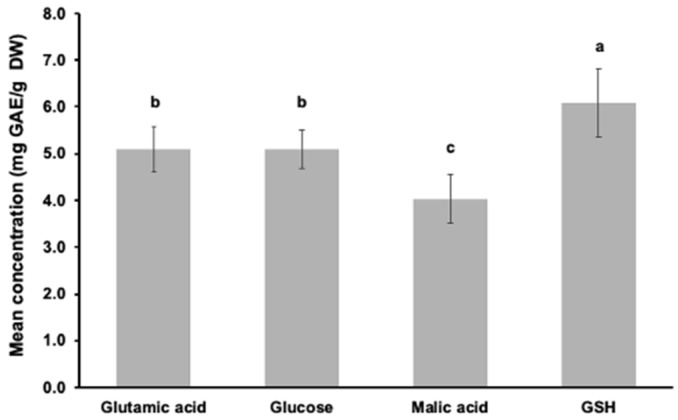
Mean reducing compound concentrations measured in the amended cultures (glutamic acid, glucose, malic and GSH) over the log phase. Different superscript letters indicate significant differences (*p* < 0.05) as determined by Wilcoxon test.

**Figure 3 biomolecules-12-00795-f003:**
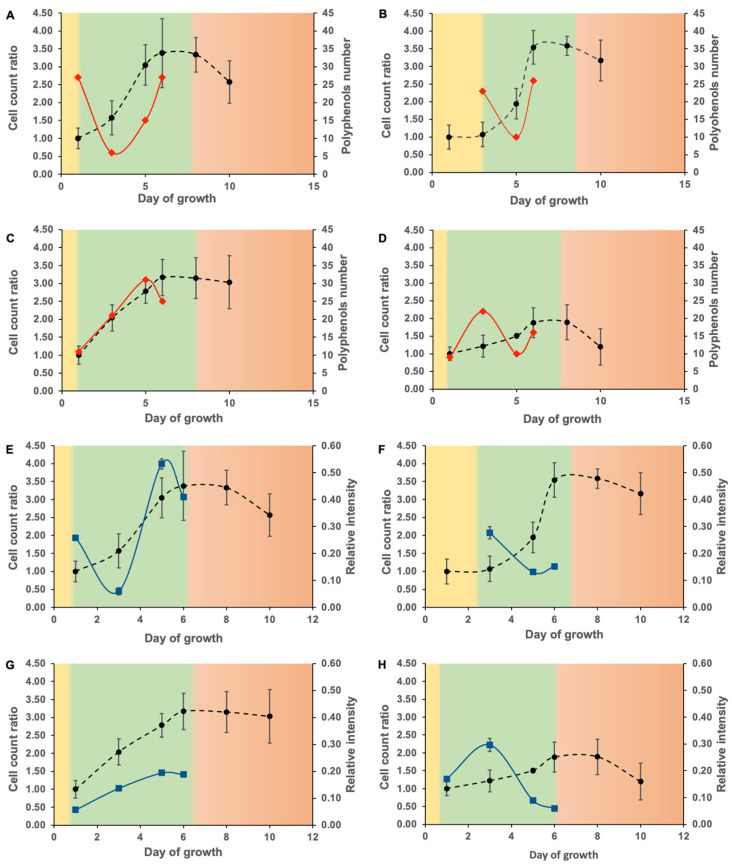
Cellular growth curve (black), number of molecular formulas (blue) and their relative intensities (red) attributed during the log phase with (**A**,**E**) glutamic acid, (**B**,**F**) glucose, (**C**,**G**) malic acid and (**D**,**H**) GSH amendment. The lag, log and stationary phases are shown in yellow, green and orange, respectively.

**Figure 4 biomolecules-12-00795-f004:**
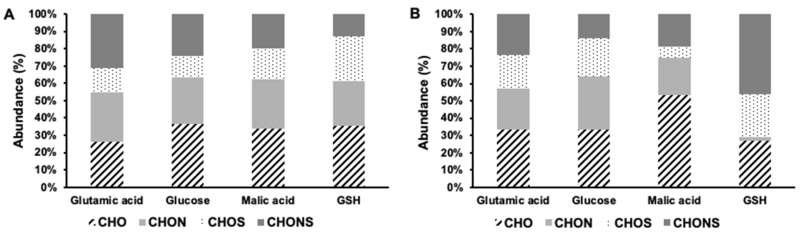
Elemental composition of phenolic compounds associated with (**A**) the minimum and (**B**) maximum growth rates. Different superscript letters indicate significant differences (*p* < 0.05) determined by Wilcoxon test.

## Data Availability

The data presented in this study are available on request from the corresponding author.

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
