# Peer review of "Influence of Carbon Sources on the Phenolic Compound Production by Euglena gracilis Using an Untargeted Metabolomic Approach"

_biomolecules, 2022, doi:10.3390/biom12060795_

Round 1

Reviewer 1 Report

the paper focus on the influence of carbon source on the production of phenolic compounds by Euglena with the purpose of using this microalga as bioremediating tool for metal contaminated waters. But the authors reserve just a glimpse to bioremediation in the introduction, since their focus seems to be only the production of chelating compounds. I would suggest testing these compounds on some of the metals commonly found in wastewater.

Figures 1-4 are too smal to be clearly readable

lux should be changed in micromoles photons

Why the cultures have been kept under photoautothropic conditions and not heterothrophic in the dark? Light is also known to inhibit glucose transport in E. gracilis, in which photoautotrophic and heterotrophic metabolic activities are reported to occur simultaneously only at very low light intensity.  

From a point of view of bioeconomy, since providing light incurs cost, the authors should test the production of chelating agents in the dark.

Reviewer 2 Report

Manuscript < INFLUENCE OF CARBON SOURCES ON THE PHENOLIC COMPOUND PRODUCTION BY EUGLENA GRACILIS USING AN UNTARGETED METABOLOMIC APPROACH >, biomolecules-1736281-peer-review-v1

In this study, the authors reported the impact of different carbon sources (i.e. glutamic acid, malic acid, glucose or reduced glutathione). in the growth medium on the phenolic production by Euglena gracilis cells. Below are some comments/suggestions for the authors’ consideration.

Main points:

In Abstract:

(1) It is recommended to supplement the key findings of this study.

(2) Line 17, Is the expression "C treatments" appropriate?  A more rigorous expression is recommended. 

(3) “metal in wastewaters” should be changed to “metals in wastewater that are”

In Introduction:

(1) It is suggested to supplement other researchers' studies on the effects of carbon sources on metabolite secretion.

(2) Some syntax errors need to be corrected (just a few are listed here).

Line 26, “ecosystem” should be changed to “ecosystems”

Line 31, “ions exchange” should be changed to “ion exchange”

……

In Result:

(1) Figure 3 suggests switching to a clear image.

(2) Line 208, Why do we measure things at such low m/z? What is m/z = 6? 

(3) In Figures 1 and 3, the serial number and title do not match.

(4) Line 234, “3C” should be changed to “3E”.

(5) Figure S5 is missing from SI.

(6) What do numbers like 1011 and 0001 mean in Figure S6?

(7) 4.1 and 4.2 subheadings are the same.

(8) Figures S5, S6 and S7 in the article are confused and do not correspond to the conclusion, so it is suggested to be checked.

(9) Some syntax errors need to be corrected (just a few are listed here).

Line 249, “each carbon sources” should be changed to “each carbon source”.

……

In Conclusions:

(1) Some syntax errors need to be corrected (just a few are listed here).

Line 304, “on the phenolic production” should be changed to “on phenolic production”.

……

Round 2

Reviewer 1 Report

I suggest the authors to focus more on bioremediation methods and bioremediation capacity of Euglena and to test the production of phenolic compounds under low light condition in their future works